# Dynamic Simulation of Permanent Magnet Synchronous Motor (PMSM) Electric Vehicle Based on Simulink

**Yunfei Zhang** [1], **Can Zhao** [2], **Bin Dai** [3] **and Zhiheng Li** [1,*]

1  Tsinghua Shenzhen International Graduate School, Tsinghua University, Shenzhen 518000, China;
   zhang-yf19@mails.tsinghua.edu.cn
2  Department of Automation, Tsinghua University, Beijing 100871, China; zhaoc17@mails.tsinghua.edu.cn
3  National Innovation Institute of Defense Technology, Beijing 100071, China; 13807318768@139.com
*  Correspondence: zhhli@mail.tsinghua.edu.cn; Tel.: +86-(153)-23831285

**Abstract:** As an important component of vehicle design and energy conservation, electric vehicle dynamics simulation is essential, especially under complicated testing conditions. The current commercial vehicle simulation software is mostly used for fuel vehicle dynamics simulation, which lacks accurate electric powertrain parts and open sources. To address this problem, this paper proposes an open-source and flexible vehicle dynamics simulation platform that includes 27 degrees of freedom (DOFs) based on Simulink, which can compatibly support both traditional vehicle dynamics simulations and electric vehicle dynamics simulations. In addition, the platform can support module customization, which is convenient for researchers. Although this platform still needs some iterations to reach industrial and commercial standards, it can already achieve parameter consistency under the stability demands in general scenarios. We believe this work should receive research attention and participation to provide lower thresholds and more references to the dynamic simulation of electric vehicles to reduce vehicle energy consumption.

**Keywords:** electric vehicle; dynamics simulation; powertrain simulation; open-source platform





## 1. Introduction

As the number of electric vehicles increases, vehicle safety is becoming increasingly important. To improve vehicle safety during driving, vehicle testing is indispensable. In actual vehicle testing, some scenarios can cause irreversible damage to the vehicle, especially under extreme conditions, which can incur high testing costs. For this reason, we use vehicle simulation methods for vehicle testing. Vehicle simulation can not only provide test results for scenarios that are difficult to test in practice but also provide an important reference for vehicle parameter optimization through large-scale simulation to reduce test costs.

At present, mainstream vehicle simulation software includes CarSim [1], ADAMS [2], AVL CRUISE [3], and VI-CarRealTime [4–6], etc. The software mentioned above have achieved good simulation results in certain simulation fields. When complex modifications are involved in a basic module, some software is not suitable for simulation demands. For example, CarSim and VI-CarRealTime can provide high-precision vehicle dynamics models, but the bottom layer of the software is not open source and does not support a customized vehicle model. ADAMS is used for multibody dynamics simulation, which is aimed at vehicle body simulation, but it is less involved in driving condition simulation. AVL CRUISE is mostly used to simulate the dynamics and economy of electric vehicles under cycling conditions, but it cannot describe the instantaneous state of motor and vehicle bodies. In addition, the above software cannot simulate electric parts, and their operation is cumbersome. To address this problem and simulate the in-wheel motors of the future, we establish an open-source and flexible vehicle dynamics simulation platform that includes the high-precision vehicle dynamics model and the powertrain model in Simulink [7,8].

The powertrain and transmission systems of electric vehicles are different from those of traditional vehicles [9]. The comparison can be described as follows:

1.  In the powertrain module, the traditional vehicle has an engine that controls the longitudinal speed by throttle percentage. The electric vehicle's powertrain is a motor that controls the speed by voltage and current of the battery;
2.  In the brake module, the electric vehicle has an energy recovery system, which is not available in traditional vehicles;
3.  In the transmission module, the traditional vehicle transmission system is generally composed of a gearbox, reducing gear, differential mechanism, and other components. In an electric vehicle transmission system, the transmission is generally integrated into the motor;
4.  In the vehicle body module, due to the low installation position of the battery, the center of gravity and the unsprung mass of electric vehicles are also different from traditional vehicles [10].

In this paper, we propose a high-precision vehicle dynamics simulation open-source platform for electric vehicles based on Simulink. In this platform, we build a vehicle model that includes a 27-DOF (degree of freedom) vehicle module and an electric powertrain module. The vehicle module includes the vehicle body model, the suspension model, the tire model, the drive model, and the brake model. which can support the simulation of the majority of scenarios in vehicle testing. In terms of the powertrain module, we design a joint simulation in the battery and motor and integrate it into the vehicle simulation platform. To simulate the transient characteristic of the motor, we use the PMSM (permanent magnet synchronous motor) model to replace the traditional mapped motor model, which can describe the state of the motor in case of failure under extreme conditions [11]. The proposed model can simulate the electric part accurately for electric vehicles, which is not complete in traditional vehicle simulation software. In addition, we maintain the traditional model, and researchers can switch models flexibly to meet different demands, which is a highlight of our platform. A graphical introduction of our simulation platform is outlined in Figure 1.

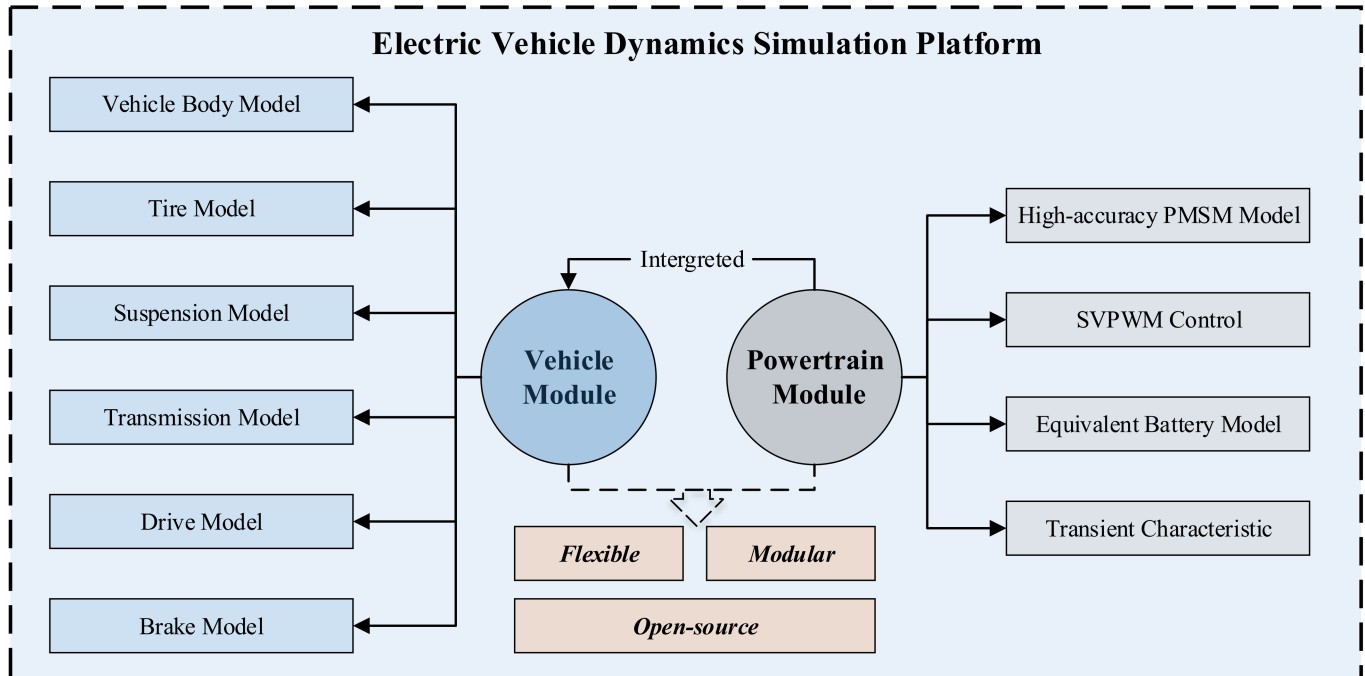

**Figure 1.** Graphical introduction of our simulation platform.

The core contribution of our work can be demonstrated as follows:

1.　We introduce an open-source simulation platform for an electric vehicle dynamics system, which addresses the difficulty of modeling the bottom layer of traditional software;
2.　The platform can provide accurate vehicle dynamics simulation and electric power-train simulation, and researchers can flexibly switch models through module selection in our platform, which is not supported in the traditional software;
3.　The platform can use a parallel server to simulate different simulation scenarios in parallel, and the results can be used for large-scale parameter optimization and vehicle stability evaluation;
4.　In future simulation design, our simulation platform can be used as a bottom layer for basic simulation, data transfer, and software interaction through the S-Function.

For reading convenience, we arrange our paper as follows. In Section 2, we introduce a 27-DOF simulation system. In Section 3, we design an electric model part including the PMSM and SVPWM models (space vector pulse width modulation). In Section 4, we present the parameters of our platform. In Section 5, we present the results and analysis of simulation scenarios and a comparison of our platform and other software.

## 2. Modeling of the 27-DOF Simulation System

In this section, we introduce an open-source vehicle dynamics simulation platform, which includes 27 DOFs: the vehicle body system, the suspension system, the tire system, the drive system, and the brake system. The degree of freedom meaning of each module can be found in Table 1.

**Table 1.** Degrees of freedom for each module.

| Module | Degree of Freedom |
| :---: | :---: |
| Vehicle body system | Lateral movement<br>Longitudinal movement<br>Vertical movement<br>Yaw rotation<br>Pitch rotation<br>Roll rotation |
| Suspension system | Suspension movement ($\times 4$) |
| Tire system | Wheel rotation ($\times 4$)<br>Tire transient characteristic ($\times 8$) |
| Drive system | Transmission |
| Brake system | Brake pressure ($\times 4$) |

In the simulation process, it is necessary to balance the speed and accuracy of the simulation. This is why we chose a model with 27 DOFs. It is worth mentioning that we use a linear transfer function in the vehicle body module and the suspension module and use a nonlinear fitting tire model in the tire module, which plays a connecting role in vehicle dynamics analysis. After comparison and analysis, we believe that the 27-DOF model is suitable for describing both the linear part and nonlinear part of the vehicle. In addition, we reserve the interface of the electric part for in-wheel motor simulation, which is introduced in detail in the following section.

At present, there are different types of models commonly used in academia and industry: the 2-DOF model [12], the 7-DOF model [13], the 22-DOF model [14], and the 27-DOF model. Different models are used in different scenarios. For example, the 2-DOF model and the 7-DOF model are typically used in theoretical analysis of vehicle stability. They can be easily modeled by ode45, which has a fast calculation speed, although the accuracy of the models is not sufficient. The 22-DOF model can better describe the dynamic parameters in vehicle stability than other models. However, the model is not suitable for electric vehicle simulation, especially in-wheel motor vehicle simulation. For the above reasons, we propose an open-source 27-DOF simulation platform, which can provide

different models in different scenarios. The basic vehicle dynamic model is shown in Figure 2.

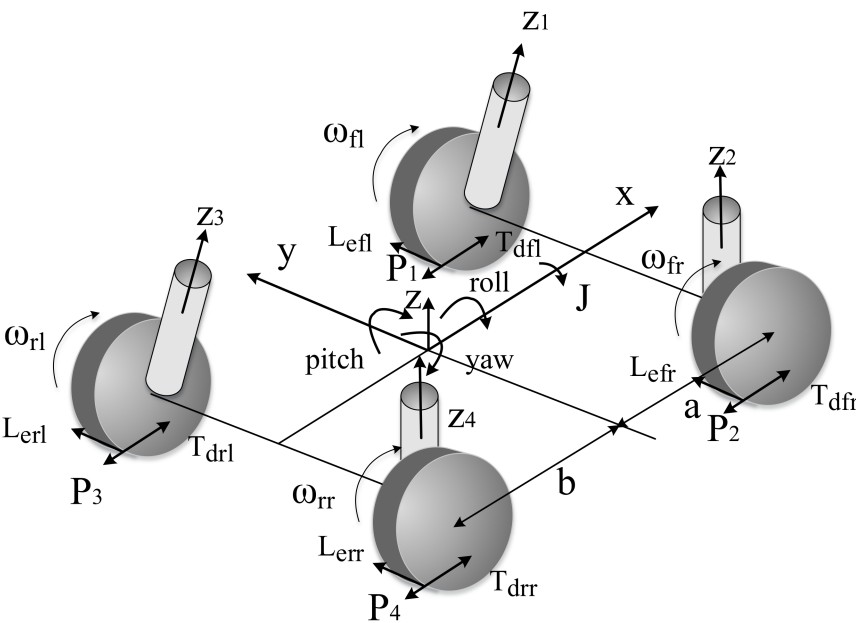

**Figure 2.** The 27-DOF model.

### 2.1. The Vehicle Body Module

In the vehicle body module, modeling using the movement and rotation of the body is a conventional method in the industry. We established a linear 6-DOF dual-track vehicle model using Simulink, which can meet the basic requirements of the simulation. The movement and rotation can be described by the following equations.

The force of the vehicle body is given by Equation (1).

$$
\begin{aligned}
F_x &= (F_{xfl} + F_{xfr})\cos\delta + F_{xrl} + F_{xrr} + (F_{yfl} + F_{yfr})\sin\delta \\
F_y &= (F_{yfl} + F_{yfr})\cos\delta + F_{yrl} + F_{yrr} + (F_{xfl} + F_{xfr})\sin\delta \\
F_z &= F_{zfl} + F_{zfr} + F_{zrl} + F_{zrr}
\end{aligned}
\tag{1}
$$

where $F_x$ represents the longitudinal force on the vehicle; $F_y$ represents the lateral force on the vehicle; and $F_z$ represents the vertical force transmitted through the suspension. $\delta$ denotes the front wheel steering angle.

The velocities of roll, pitch, and yaw are given by the following equation.

$$
\begin{bmatrix} p \\ q \\ r \end{bmatrix} =
\begin{bmatrix}
1 & \sin\phi\tan\theta & \cos\phi\tan\theta \\
0 & \cos\phi & -\sin\phi \\
0 & \frac{\sin\phi}{\cos\theta} & \frac{\cos\phi}{\cos\theta}
\end{bmatrix}^{-1}
\begin{bmatrix} \dot{\phi} \\ \dot{\theta} \\ \dot{\psi} \end{bmatrix}
\tag{2}
$$

where $[p\ q\ r]^T$ represents the body-fixed angular velocity vector; $[\phi\ \theta\ \psi]^T$ represents the Euler angles, which are the roll angle, pitch angle, and yaw angle, respectively; and $[\dot{\phi}\ \dot{\theta}\ \dot{\psi}]^T$ is the rate of change of the Euler angles.

### 2.2. The Suspension Module

The suspension degree of freedom equation is given as follows.

$$
F_{z1} = K_{t1}(z_{g1} - z_{\omega 1}) + K_1(z_{b1} - z_{\omega 1}) + C_1(\dot{z}_{b1} - \dot{z}_{\omega 1})
\tag{3}
$$

The equation represents one suspension movement on the z-axis, and the rest of the three suspension models are similar. $F_{z1}$ is the vertical force for one suspension; $K_{t1}$ is the tire stiffness coefficient; $z_{g1}$ is the vertical input coefficient of the ground; $z_{b1}$ is the movement of the vehicle body; $K_1$ is the kinematic coefficient of suspension; and $C_1$ is the compliance coefficient of suspension.

### 2.3. The Tire Module

In a vehicle dynamics analysis, the tire model plays an important connecting role in a simulation. The input parameters include the vehicle body parameters, driving parameters, and brake parameters, which directly affect the tire force. The tire force acts on the suspension and the vehicle body in turn, which forms closed-loop control. Therefore, a suitable tire model can obviously improve the speed and accuracy of the simulation.

We have investigated a variety of tire models. The currently widely used models in the industry include the Fiala model [15], the Lugre model [16], the Unitire model [17], and the magic formula model [18]. The Fiala model makes simple approximate assumptions and is generally used in the stability analysis of bicycle models. The Lugre model performs complex calculations through partial differential equations, which takes a long time in simulation. The Unitire model and the magic formula model are similar. The Unitire model is a semiempirical model that can be modeled with a small amount of data, whereas the magic formula model is fitted with a large amount of experimental data. When the data are abundant, the magic formula model is preferred. Here, we chose a magic formula model to describe the nonlinear part, which can perform better than other models in our simulation. In addition, researchers can flexibly switch to other models. The equation for the magic formula model is given by Equation (4).

$$y(x) = D \sin[\text{Carctan}\{Bx - E(Bx - \arctan(Bx))\}] \tag{4}$$

where $y(x)$ can describe the longitudinal slip and the side slip characteristics.

To describe the transient characteristic of the tire, the relaxation length can be described as follows.

$$T_d(s) = \frac{1}{|\omega| R_e s / L_e + 1}(F_x R_e + M_y) \tag{5}$$

where $T_d$ is the combined tire torque; $\omega$ is the wheel angular velocity; $R_e$ represents the effective tire radius; and $L_e$ denotes the tire relaxation length.

### 2.4. The Drive Module

In the drive module, different vehicles have different requirements in simulation. Considering the simulation with in-wheel motor electric vehicles, we established different drive models, and researchers can flexibly switch models in simulation, which includes the mapped engine model, the mapped motor model, the PMSM model, and the Ansys motor model. The models are introduced in detail in the subsequent section. In addition, the transmission equation is given in Equation (6).

$$Jk\frac{d\omega_m}{dt} = T_e - T_L - B_1 k\omega_m \tag{6}$$

where $J$ represents the motor moment of inertia; $\omega_m$ is the rotation speed of the motor; $T_e$ is the output torque; $T_L$ is the load torque; $B_1$ is the damping coefficient; and $k$ is the gear ratio.

### 2.5. The Brake Module

The brake module can be found in our previous research [19], which uses a serial mapped control method to establish an energy recovery system. The equation is given below.

$$[T, P_r] = f(P, \omega) \tag{7}$$

where $T$ is the braking torque; $P_r$ is the regenerative power; $P$ represents the braking pressure; and $\omega$ is the tire rotating speed.

## 3. Modeling of the Powertrain Part

In the powertrain model selection, the powertrain model [20] can be divided into the following types: a mapped engine model, a mapped motor model [21], a PMSM model, and an Ansys motor model. In the actual testing, we found that the mapped model is too rough to represent the transient characteristic of the powertrain, which can be proven in the results. However, the Ansys model is too complex and caused the simulation time to increase exponentially, which cannot meet the demands of real-time simulation. The PMSM empirical formula model could not only express the characteristics of the motor in the case of instantaneous acceleration and deceleration but could also flexibly select the step size to meet the needs of real-time simulation. Therefore, we established a detailed PMSM model to meet the needs of in-wheel motor electric vehicle simulation. In addition, we maintained the interface of the models mentioned above to facilitate research. The physical model here used some components in the Simscape toolbox of Simulink, and we have reconstructed the model and the compatibility of some key components, which can be found in the Supplementary Materials.

To ensure the normal operation of the system and the compatible simulation with the platform, we arranged the motor model and its control circuit into several parts: the current PI model [22]; the speed PI model [23]; the Clark transform model [24]; the park transform model [25]; the anti-park transform model [26]; the SVPWM model [27]; the IGBT (Insulated Gate Bipolar Transistor) model [28]; the PMSM model [29]; and the measurement model.

In the PMSM model, the electric dynamics can be expressed as follows.

$$\begin{aligned} \frac{di_{sd}}{dt} &= \frac{u_{sd}}{L_d} + \frac{L_q}{L_d}p\omega_m i_{sq} - \frac{R_s}{L_d}i_{sd} \\ \frac{di_{sq}}{dt} &= \frac{u_{sq}}{L_q} - \frac{L_d}{L_q}p\omega_m i_{sd} - \frac{R_s}{L_q}i_{sq} - \frac{\psi_{pm}}{L_q}p\omega_m \\ \frac{d\varphi_m}{dt} &= \omega_m \end{aligned} \tag{8}$$

where $u_{sd}$ and $u_{sq}$ are the stator voltages in the d-axle and q-axle, respectively; $i_{sd}$ and $i_{sq}$ are the stator currents in the d-axle and q-axle, respectively; $R_s$ is the stator resistance; $L_d$ and $L_q$ are the equivalent inductances in the d-axle and q-axle, respectively; $\psi_{pm}$ is the electromotive force; $p$ is the number of pole pairs; and $\omega_m$ is the mechanical spin speed of the rotor.

The output torque of the PMSM model can be described as

$$T_e = \frac{3}{2}p[\psi_{pm}i_{sq} + (L_d - L_q)i_{sd}i_{sq}] \tag{9}$$

where $T_e$ is the output torque of the PMSM model.

In the current PI closed-loop model and the speed PI closed-loop model we use the $i_d = 0$ control method [30] to achieve good speed tracking performance. The equation is as follows.

$$T_e^* = K_p(n^* - n) + K_i \int (n^* - n)dt \tag{10}$$

where $K_p$ represents the proportional coefficient; $K_i$ represents the integral coefficient; $T_e$ means output torque; $n$ is motor speed; and * denotes that the variable is an update value or measurement value.

There are many methods in the motor control strategy, such as MTPA [31], MTPV [32], and RST [33], etc. Here we chose a simplified control method for model building, and researchers can make modifications on this basis.

The theories for the Clark transform model, the park transform model, the anti-park transform model, and the SVPWM model can be found in references [22–29]. The IGBT model can be found in the Simscape database. We modified the PMSM model to receive load torque and customize the parameters to make it suitable for electric vehicle simulation.

The measurement model can output motor torque, motor rotation angle, current, and motor rotation speed.

The model mentioned above can be arranged as follows in Figure 3. The input parameters include the speed command and torque command, which are the initial states of the PMSM model.

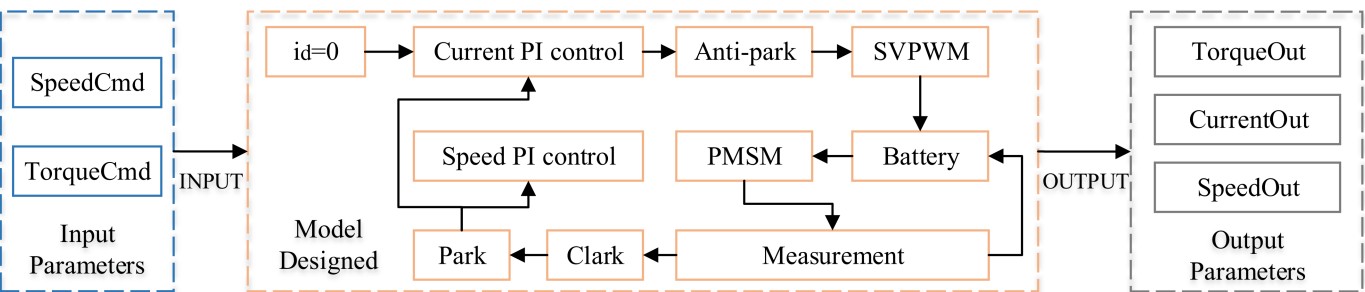

**Figure 3.** The PMSM model.

## 4. Simulation Parameters and Scenarios

The platform includes the vehicle, motor, and scenario parameters. The table shows each parameter's name, abbreviation, unit, and default value. The default values are based on the existing E-class Sedan vehicle database [34] and the Simscape database [35]. To adapt the electric vehicle simulation, parameters V1–V10 were specified, such as the sprung mass on one suspension (V8) and the transmission ratio (V9). Parameters M1-M9 account for the PMSM motor model and the battery model, which were derived from the default three-phase PMSM model in Simscape. To validate the effectiveness of our simulation platform, we used the const radius road scenario [36] and the double lane change scenario [37] for simulation tests, which can be found in Figures 4 and 5. Scenario parameters S1–S3 are given below and can be modified by users. In addition, similar results will be obtained even if the parameters are changed.

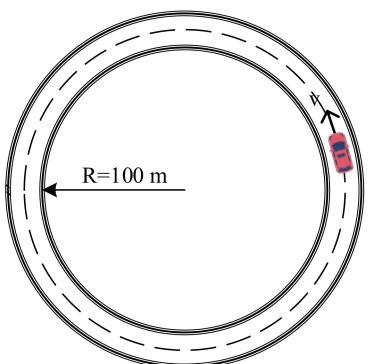

**Figure 4.** Const Radius Road Scenario.

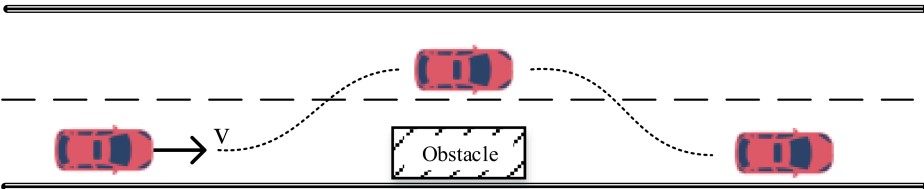

**Figure 5.** Double Lane Change Scenario.

In terms of the remaining models, such as the tire model and the suspension model, a default model can satisfy the simulation demand. In the tire model, the commonly used

model is the magic formula model, which can provide fitting parameters from measurement. The electric vehicle suspension design usually uses the Macpherson independent suspension [38] for the front axle and the solid suspension [39] for the rear axle, which can be found in the Simulink database.

The simulation parameters are shown in Table 2.

**Table 2.** Simulation Parameters.

| No. | Parameter | Abbr. | Unit | Value |
| --- | --- | --- | --- | --- |
| V1 | Mass of vehicle | $m$ | kg | 1100 |
| V2 | Distance from center to front axle | $a$ | m | 1.50 |
| V3 | Distance from center to rear axle | $b$ | m | 1.50 |
| V4 | Height form center to axle | $h$ | m | 0.1 |
| V5 | Inertia of roll | $I_x$ | kg·m$^2$ | 1922 |
| V6 | Inertia of pitch | $I_y$ | kg·m$^2$ | 432 |
| V7 | Inertia of yaw | $I_z$ | kg·m$^2$ | 2066 |
| V8 | Sprung mass on one suspension | $m_1$ | kg | 200 |
| V9 | Transmission ratio | $k$ | 1 | (3.1, 1) |
| V10 | Max brake pressure | $P$ | pa | 8e6 |
| M1 | Output torque of the motor | $T_e$ | Nm | 150 |
| M2 | Battery power | $U_e$ | V | 311 |
| M3 | Motor speed | $n$ | rpm | 2000 |
| M4 | Equivalent inductance (d) | $L_d$ | H | 0.0006 |
| M5 | Equivalent inductance (q) | $L_q$ | H | 0.0006 |
| M6 | Equivalent resistance | $R_s$ | Ohm | 0.05 |
| M7 | Inertia of motor | $J$ | kg·m$^2$ | 0.011 |
| M8 | Damping coefficient | $B$ | 1 | 0.002 |
| M9 | Pole pairs | $p$ | 1 | 4 |
| S1 | Speed max | $V_m$ | km/h | 90 |
| S2 | Radius | $R$ | m | 50,100 |
| S3 | Friction coefficient | $\mu$ | 1 | 0.85 |

## 5. Results and Analysis

In the actual testing process, the conditions of some extreme scenarios are harsh, which easily causes irreversible damage to the experimental equipment and induces a high testing cost. To reduce testing costs, the high-accuracy commercial simulation software CarSim is typically used as a control group in the analysis process. In the simulation results, the common evaluation indicators are yaw velocity, side slip angle, and front-wheel steering angle. In the const radius scenario and the double lane change scenario, the yaw velocity and the trajectory are suitable for simulation quality analysis.

In terms of the electric powertrain system, the traditional mapped motor model does not consider the actual scenario. To simulate the transient electric characteristic, we compared the mapped motor and the accurate PMSM model, which can consider the efficiency and the power limitation. The results demonstrate the difference between the mapped motor model and the proposed electric powertrain system.

### 5.1. Simulation in the Const Radius Scenario

To verify the effectiveness of the proposed simulation platform, we compared the 27-DOF model and CarSim in the const radius scenario. In vehicle stability evaluation, the yaw velocity is typically used for analysis. However, simple curve analysis cannot fully explain the effectiveness of the simulation platform. Therefore, we compared the vehicle trajectory and compared the different models in different scenarios to complement it.

The yaw velocity in the proposed model and CarSim is shown in Figure 6.

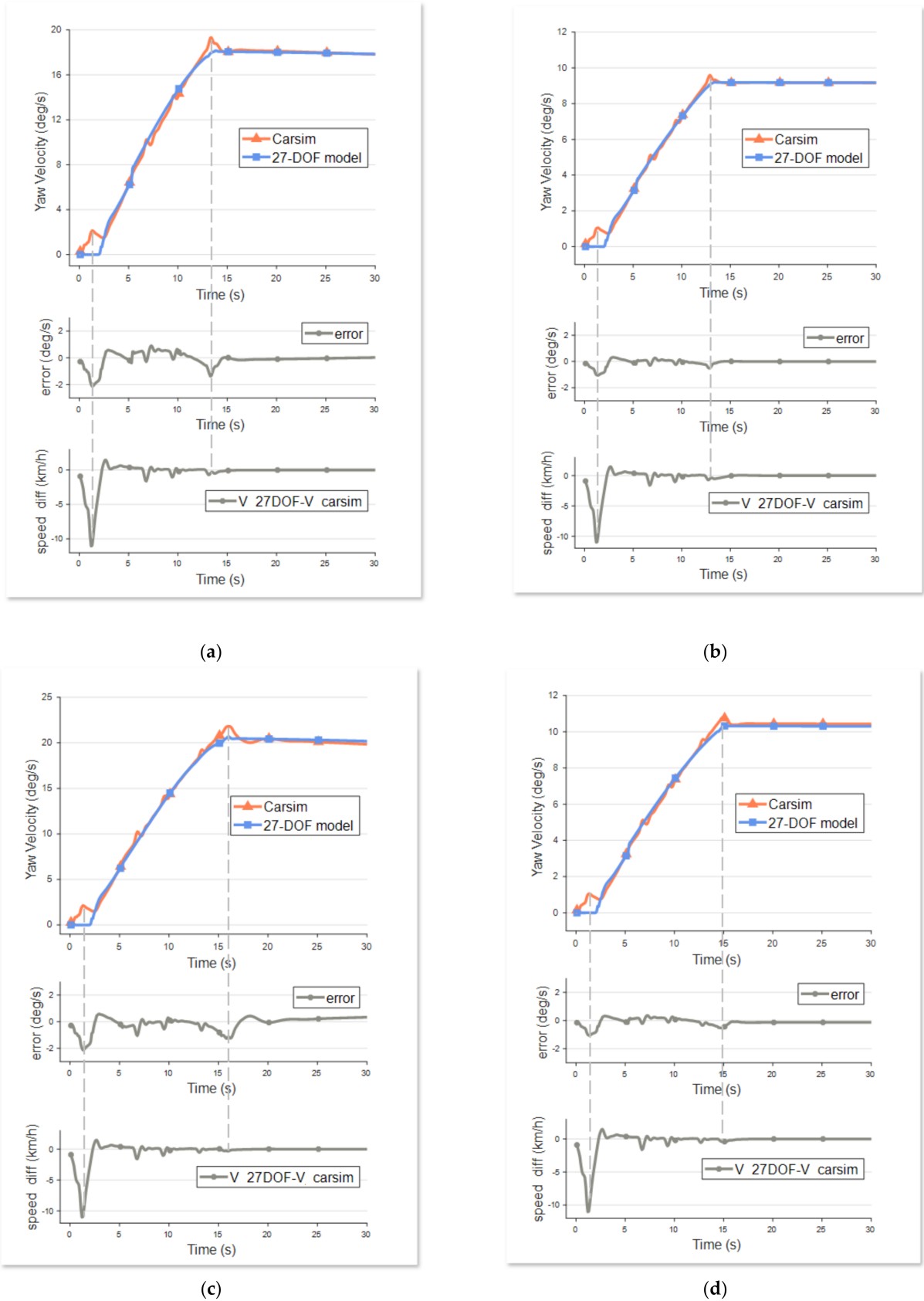

**Figure 6.** Yaw velocity of the 27-DOF model and CarSim in different scenarios. (**a**) speed = 55 km/h; radius = 50 m. (**b**) speed = 65 km/h; radius = 50 m. (**c**) speed = 55 km/h; radius = 100 m. (**d**) speed = 65 km/h; radius = 100 m.

We analyzed the two main parameters of vehicle speed and turning radius under const radius condition, and gave four scenarios under different combinations, which could represent the environmental parameters from normal driving conditions to extreme conditions. From the acceleration state to the steady state, the stability parameters of each scene showed good followability, especially in the steady state stage, the simulation error of the two platforms were close to zero, which illustrates the effectiveness of our simulation platform.

In Figure 6, we compare the yaw velocity of the proposed 27-DOF model and CarSim in different scenarios. In the period of (0 s, 5 s), the yaw velocity of the CarSim model started vibrating, which was caused by the different speed control and steering control in different software. In this span, the speed was approximately equal to zero, which had little effect on vehicle stability. In the time period of (10 s, 15 s), the CarSim result showed little vibration and little overshoot, which was caused by the path tracking control of the steering system.

The results demonstrate that the two models are consistent, and our simulation platform can work well under this condition. In addition, the vibration and the overshoot of our model perform better than CarSim (CarSim2019, Mechanical Simulation Corporation, USA), which is caused by linear approximation simplifications in Section 2. This platform can meet our simulation requirements for accuracy and efficiency.

The vehicle position in the proposed model and CarSim in the scenario of speed = 55 km/h and radius = 100 m is shown in Figure 7.

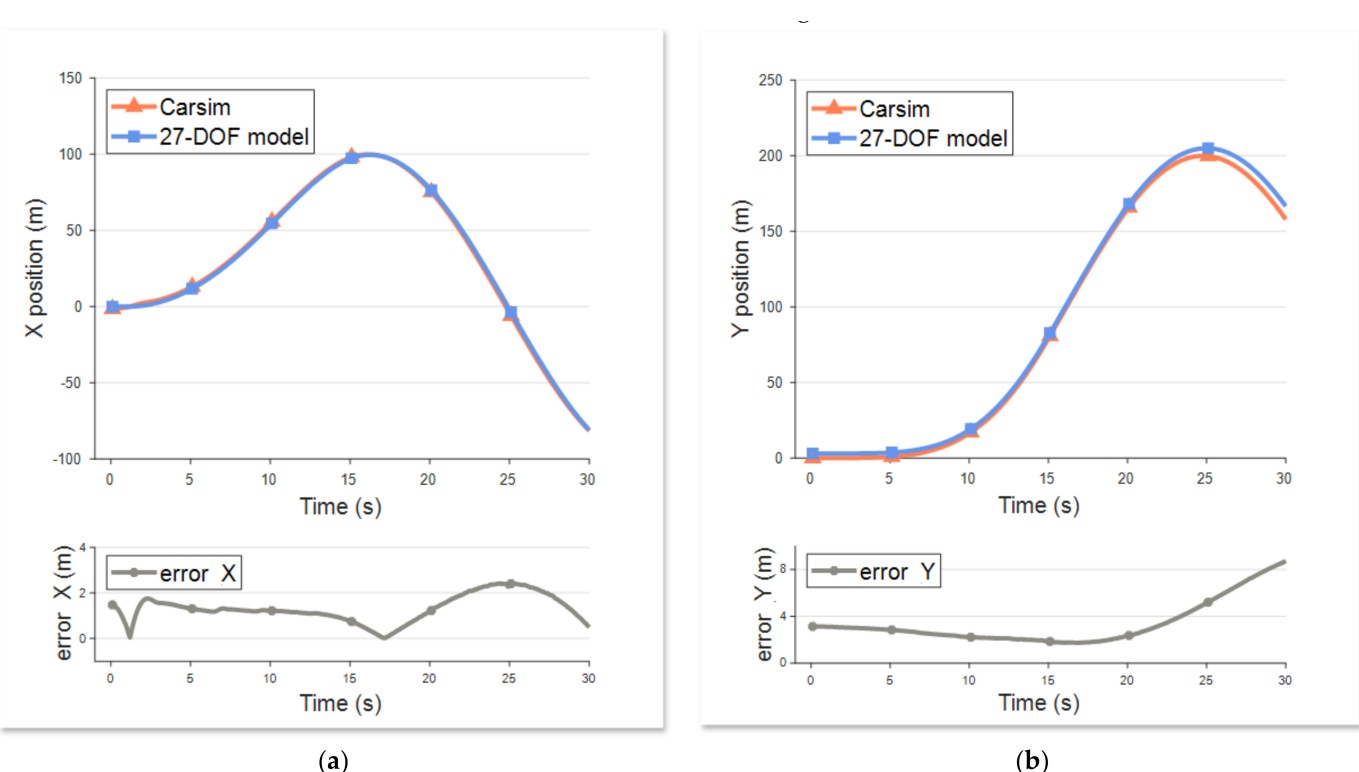

(**a**)                                                               (**b**)

**Figure 7.** Vehicle position of the 27-DOF model and CarSim in the Const Radius Scenario of speed = 55 km/h and radius = 100 m. (**a**) X Position. (**b**) Y Position.

In Figure 7, we show the vehicle position of the 27-DOF model and CarSim in the const radius scenario. The result demonstrates a semicircle of the constant radius scenario, which can represent the vehicle state in the acceleration and stabilization phases. At (15 s, 30 s), the error increases, which is caused by the different speed controls at the stable state. In the speed input part, we input the same target speed sequence into the two compared platforms. During the simulation process, we found that the proposed model could follow the speed command, but CarSim showed accumulated errors, which caused a large error in

Y position. The results of the vehicle position can supplement the usability of our platform in this situation.

Consequently, we compared different parameters in the const radius scenario with different models, and the errors can be found in Tables 3 and 4. Notably, our model appears best in all models in the analysis of relative errors in different scenarios. Analogously, in the stable state of the simulation, our model is close to the CarSim model, which demonstrates that our simulation platform is effective in the const radius scenario. In the 2-DOF model and the 7-DOF model, the relative error and the stable error are the largest, which is caused by the linear model simplification and the linear tire force. In the 22-DOF model, the result is close to CarSim, which is due to the addition of a nonlinear tire model. The stable error in the 27-DOF model was the lowest as a result of the good speed control in powertrain simulation. The module mentioned above can be selected by users in individuation in our platform.

**Table 3.** Yaw velocity relative error in different scenarios of const radius.

| Scenarios | Speed(km/h) | 55 | 65 | 55 | 65 |
| --- | --- | --- | --- | --- | --- |
| | Radius (m) | 100 | 100 | 50 | 50 |
| Models | 2-DOF | 16.2% | 17.54% | 18.02% | 20.29% |
| | 7-DOF | 11.09% | 12.12% | 12.53% | 15.81% |
| | 22-DOF | 9.87% | 10.23% | 10.63% | 12.73% |
| | **27-DOF** | **7.70%** | **8.31%** | **9.19%** | **11.25%** |

**Table 4.** Yaw velocity stable error in different scenarios of const radius.

| Scenarios | Speed(km/h) | 55 | 65 | 55 | 65 |
| --- | --- | --- | --- | --- | --- |
| | Radius (m) | 100 | 100 | 50 | 50 |
| Models | 2-DOF | 14.6% | 15.7% | 14.6% | 15.7% |
| | 7-DOF | 2.49% | 2.23% | 2.56% | 2.73% |
| | 22-DOF | 2.49% | 2.23% | 2.56% | 2.73% |
| | **27-DOF** | **0.39%** | **0.34%** | **0.03%** | **0.13%** |

*5.2. Simulation in the Double Lane Change Scenario*

In the previous section, we demonstrated that our simulation method is effective in constant radius conditions. To clarify the universality of our model, we also simulated a typical scenario of a double lane change. The double lane change scenario covers the condition of turning and turning back, which is a commonly used scenario for evaluating vehicle stability. The result of yaw velocity can be found in Figure 8.

As shown in Figure 8, we tested the turning stability in the double lane change scenario of unsaturated steering and saturated steering. Similar to const radius condition, we selected two scenarios of vehicle speed from the normal state to the extreme state. The vehicle speed of the unsaturated steering condition was 55 km/h and the vehicle speed of the saturated steering condition was 65 km/h. At time $t_{in}$, the vehicle enters the side lane, and at time $t_{out}$, the vehicle returns to the original lane. In this period, the vehicle changes lanes between the two lanes. At time $t_{peak}$, the yaw velocity reaches the maximum value, and the error is within the controllable range, which verifies the effectiveness of our platform in the double lane change scenario of both unsaturated steering and saturated steering.

The results demonstrated that the vehicle state can be controlled in a normal range, and the yaw velocity of our platform is consistent with CarSim.

The vehicle position in the proposed model and CarSim is shown in Figure 9.

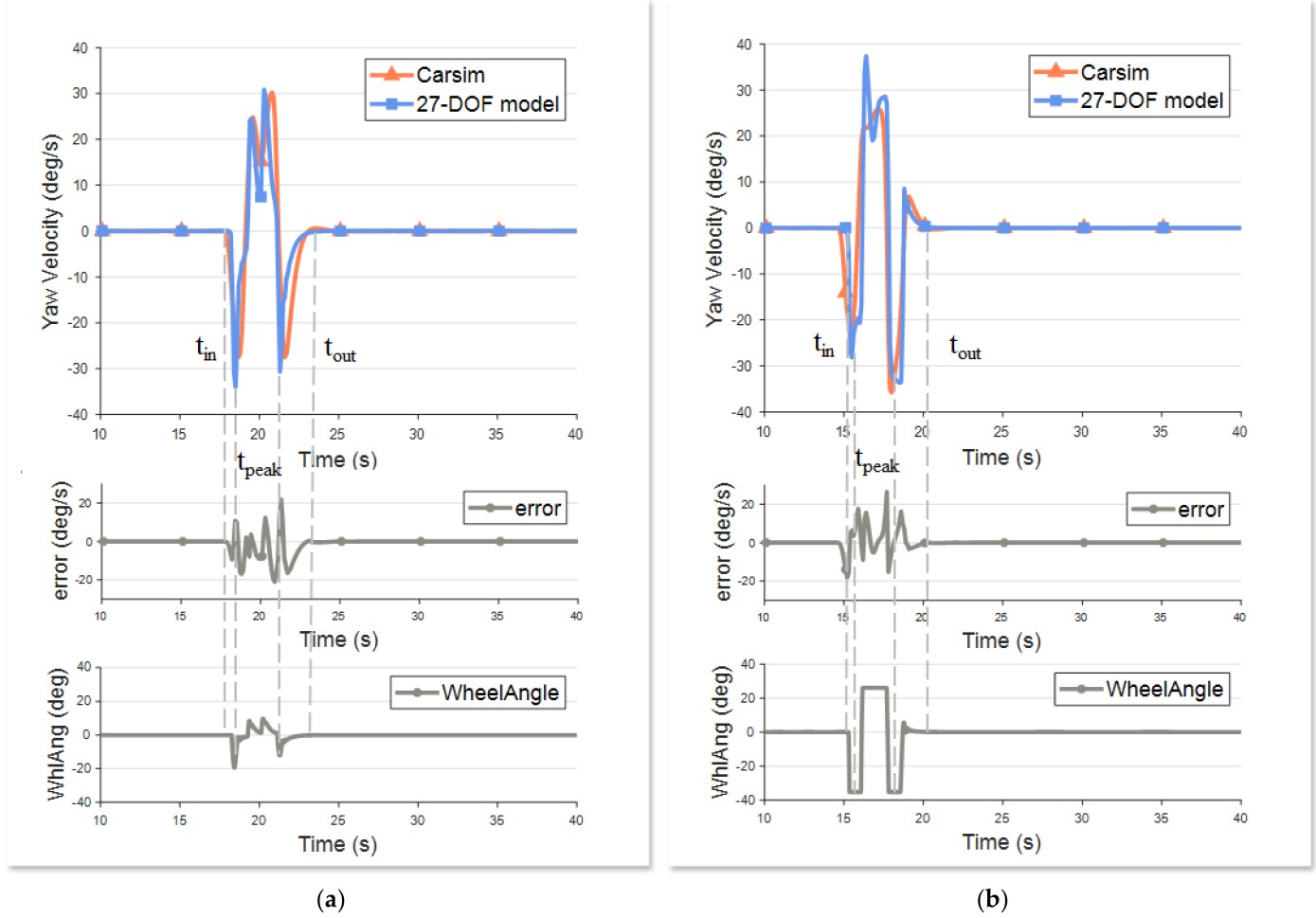

**Figure 8.** Yaw Velocity in the double lane change scenario of unsaturated steering and saturated steering. (**a**) Unsaturated Steering. (**b**) Saturated Steering.

Shown in Figure 9, similar to the const radius condition, we show the vehicle position of the 27-DOF model and CarSim in the double lane change scenario of the unsaturated steering condition. The result demonstrates the vehicle state in the steering and turn back phases. At 25 s, the error reached the maximum value, which was caused by the different speed controls at the brake state. In the proposed model, the vehicle longitudinal speed slowly decreased to zero after turning back to the original lane. In the CarSim model, the vehicle longitudinal speed suddenly decreased to zero after turning back to the original lane due to a large brake pressure. This created the accumulated error of the X position. However, our region of interest is in the turning period, and the maximum value of the X position error is not included in this period, which will not affect our results. The Y position error was approximately zero, which ensured consistency of control in the lateral direction.

Analogously, the results of the vehicle position can supplement the usability of our platform in this situation. Consequently, the results of our platform simulation in the double lane change scenario showed good path tracking performance and good stability parameter calculation.

*5.3. Simulation in the PMSM Model*

During the driving process of the vehicle, the rotation speed response is often not instantaneous. It is related to the response state of the motor and the output power of the battery. To validate the effectiveness of the PMSM model, we compared the command torque and the command rotation speed of the PMSM model and the mapped motor model. The results are shown in Figure 9.

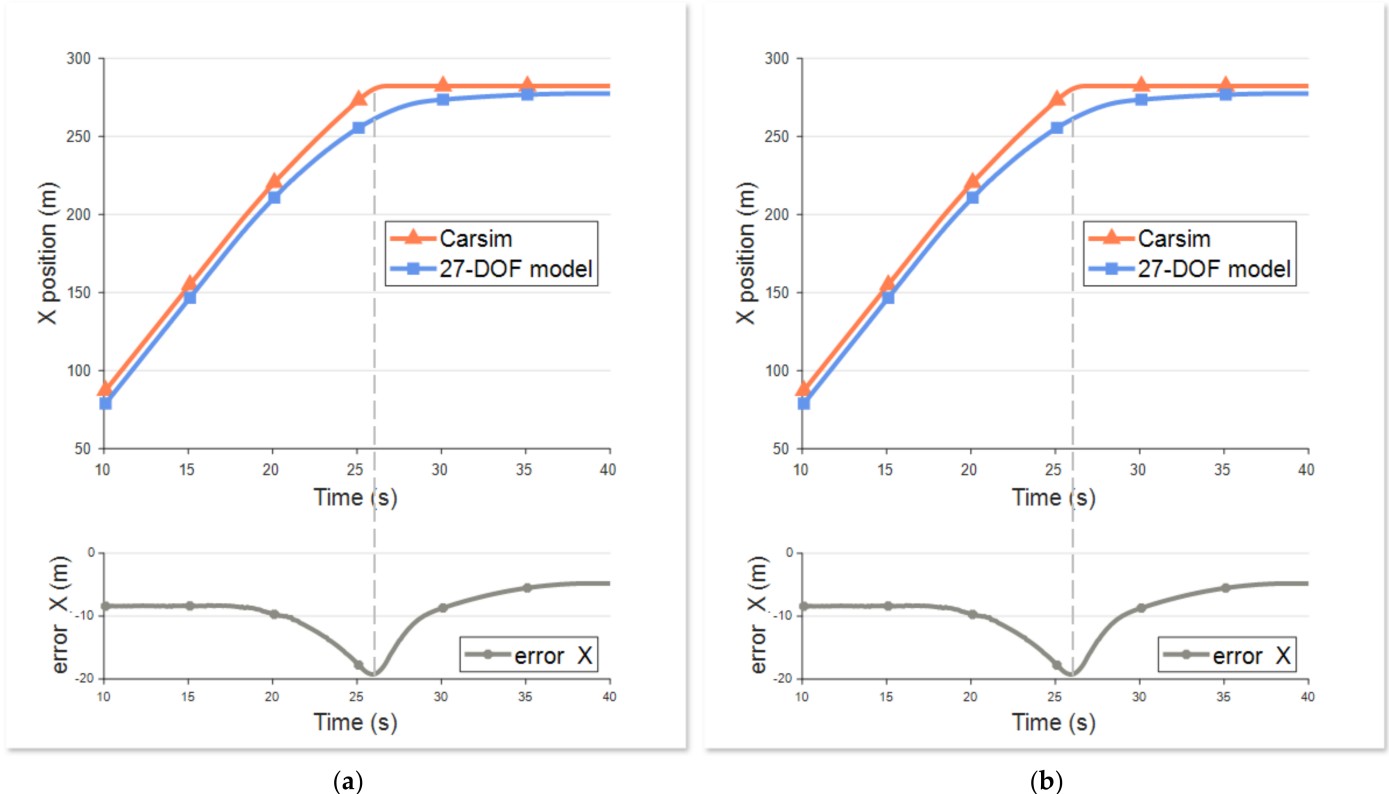

**Figure 9.** Vehicle position of the 27-DOF model and CarSim in the double lane change scenario of unsaturated steering condition. (**a**) X Position. (**b**) Y Position.

In Figure 10a, we compared the output torque of the mapped motor model and the PMSM model. The system ran at 1 s, and the torque command tureds to 150 Nm, which led to an overshoot value in motor output. In the period of (2.5 s, 7.5 s), the vehicle speed increased, and the mapped motor model and the PMSM model stabilized at 120 Nm. After that, the speed reached the preset value, and the torque decreased. In the period of (8 s, 20 s), the vehicle speed was stable, while the mapped motor model and the PMSM model decreased to 50 Nm.

In Figure 10b, we compared the output rotation speed between the mapped motor model and the PMSM model. The curve demonstrates that the mapped motor model rotation speed reached 4000 rpm, while the PMSM model rotation speed was maintained at 2000 rpm at a period of (5 s, 8 s). Similar to the torque result, the first peak value was caused by the system start. The result also indicates the transient characteristic of the motor. At (7.5 s, 9 s), the rotation speed showed a temporary increase, which was caused by the sudden decrease in motor torque.

In the comparison of the PMSM empirical model and the mapped motor model, the mapped motor model is ideal, which can provide the command value within the allowable parameter range, whereas the PMSM model showed little vibration, which is close to reality. In addition, the results indicate that the mapped motor model considered fewer efficiency issues, whereas the PMSM model not only considered the efficiency issues but also considered the motor power provided by the battery model.

After comparing the results of the mapped motor model and the PMSM empirical model, we prefer to use the PMSM model for simulation, which can achieve better results. The effect of the powertrain part on vehicle dynamics is mainly related to the speed following results, which can be found in the yaw velocity results of the aforementioned const radius scenarios in Section 5.1. The powertrain model used in the const radius scenario of the proposed model is the PMSM empirical model, and the powertrain model

used in CarSim is a mapped engine model. Compared with CarSim, the longitudinal speed follows better in the proposed model, which is also an advantage of our proposed platform.

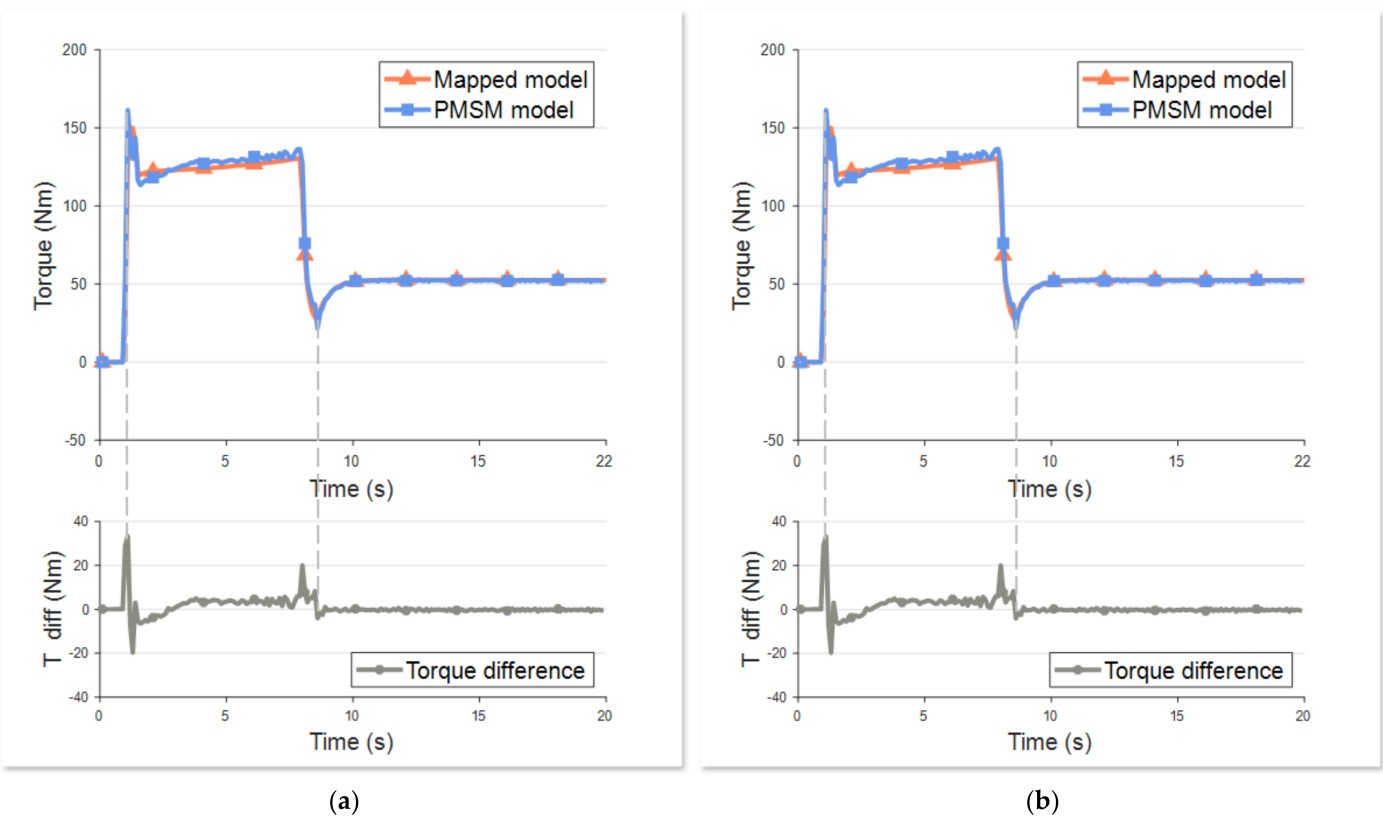

(**a**)                                    (**b**)

**Figure 10.** Torque and Rotation Speed in Different Models. (**a**) Torque. (**b**) Rotation Speed.

## 6. Conclusions

In this paper, we propose an accurate open-source electric vehicle dynamics simulation platform, which provides a 27-DOF vehicle simulation and an electric powertrain simulation. Our main goal was to innovatively establish an integral electric vehicle model and test the effectiveness in the constant radius conditions and in the double lane change conditions of the platform. In the vehicle dynamics modeling part, we validated that the accuracy of our platform is the highest; in addition, in the electric modeling part, we carried out high-precision modeling of the battery and motor. We believe that the proposed platform can work well under most conditions. We believe this paper will attract research attention for use of our platform as a foundation to achieve more results.

However, there exists some space for improvement in our proposed system. In some specific conditions, our system may have some limitations. In future work, we will iteratively upgrade the built platform and focus on the application of parameter identification [40] to improve the versatility and efficiency of simulation through deep learning [41].

**Supplementary Materials:** The physical model files are available online at https://github.com/1332 231041/electric-vehicle-dynamics-simulation.

**Author Contributions:** Formal analysis, Y.Z.; Methodology, Y.Z.; Supervision, B.D. and Z.L.; Validation, Y.Z.; Writing—original draft, Y.Z.; Writing—review & editing, C.Z. and Z.L. All authors have read and agreed to the published version of the manuscript.

**Funding:** This work was supported by the National Natural Science Foundation of China (61790565), and Science and Technology Innovation Committee of Shenzhen (CJGJZD20200617102801005).

**Institutional Review Board Statement:** Not applicable.

**Informed Consent Statement:** Informed consent was obtained from all subjects involved in the study.

**Data Availability Statement:** Data is available from the first author upon request.

**Conflicts of Interest:** The authors declare no conflict of interest.

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
