# Peer review of "Dynamic Simulation of Permanent Magnet Synchronous Motor (PMSM) Electric Vehicle Based on Simulink"

_energies, doi:10.3390/en15031134_

Round 1

Reviewer 1 Report

The authors must provide a physical model of their Powertrain Part.

The control system is not presented (which control loop has been developed, speed control, current control etc.?)

In the modeling and simulation of electric vehicle dynamics, several parameters are not taken into account, among others: the road situation, the vehicle mass, etc.

The authors should develop part 4 in their study.

Figures No. 6,7,8,9 and 10 need further analysis and explanation.

The vehicle position of the 27-DOF model is not well developed.

The references are poor, this part must be improved.

The authors can use the references below to improve their article.

  • A cyber physical energy system design (CPESD) for electric vehicle applications, 12th System of Systems Engineering Conference (SoSE),2017.
  • Vehicle Dynamic Simulation for Efficiency Optimization of Four-wheel Independent Driven Electric Vehicle, World Electric Vehicle Journal, 2010.
  • Robust RST controller design for induction motor drive for electric vehicle application, International Conference on Green Energy ICGE 2014.

Author Response

Thanks a lot for raising these questions!

The response can be found in the attached file.

Reviewer 2 Report

The paper is presents a software for simulation of coupled dynamic behaviour of an electric vehicle.

The main comment is about the scientific novelty of the work which seems to be lacking. This work can be performed using the Simscape toolbox of Matlab/Simulink, which does not include a scientific novelty.

1. What is the main question addressed by the research?

The main question is the development of open-source Simulink based software used to study electric vehicle dynamics (longitudinal, vertical and lateral).

2. Do you consider the topic original or relevant to the field? Does it address a specific gap in the field?

The topic is original in the sense that carrying an idea to develop an open-source software which researchers can modify and share the new developed modules through S-functions. The authors mention that 27-DOF model is missing in the literature and describe the model in the paper.

3. What does it add to the subject area compared with other published material?

The paper presents a 27-DOF model of the dynamics of the vehicle with electric drivetrain. The models exist in the literature separately, but coupling the models represents a novel part of the paper.

4. What specific improvements should the authors consider regarding the methodology? What further controls should be considered?
Would be better to explain what is advantage with respect to using "Vehicle Dynamics Blockset" of Simulink. As in both cases Matlab/Simulink is required.
The authors provide set of separate comparative simulations as a validation of the simulator. Which demonstrates the validity of the model in module-wise. However, as the main advantage of the simulator is that it accounts all 27-DOF, it would be reasonable to see when all the modules are active. For example, stability of the electric vehicle on a uneven/bumpy road with some curvature (or in double lane change scenario).

5. Are the conclusions consistent with the evidence and arguments presented and do they address the main question posed?
The conclusions are consistent with current content of manuscript.

6. Are the references appropriate?
The reference are appropriate.

7. Please include any additional comments on the tables and figures.
The lines in the figures could have different line styles. 

Author Response

Thanks a lot for raising this question!

The response can be found in the attached file.

Reviewer 3 Report

Just some minor errors to be corrected:

pg. 1, row 33: "have achieved" instead of "has achieved"

pg. 6, row 183: "...engine model...motor model..." - which is the difference between these two and to what type of engine/motor are you referring, different from the electric motor - PMSP ?

a) The paper presents a new 27 DOF model of an electric vehicle, combining previously developed modules and new ones. However, the title of the paper is referring to "vehicle dynamics", which would lead the potential reader to think that the acceleration performance is studied. Despite this, the scenarios taken into account (constant radius road and lane change) provided results regarding the vehicle stability, in terms of yaw rate. So, my opinion is that either the title of the paper should be changed in order to reflect the contents or some results regarding the vehicle dynamics, based on the PMSM

b) For the lane change scenario the speed of the vehicle should be mentioned.

c) Table 2 shows that the vehicle has 2 transmission ratios; which one was considered for the simulations?

d) pg. 1, row 33: "have achieved" instead of "has achieved"!

e) pg. 6, row 183: "...engine model...motor model..." - which is the difference between these two and to what type of engine/motor are you referring, different from the electric motor - PMSP ?

f) referring to the PMSM model: the electric engine dictates the dynamics of the vehicle...maybe the authors would consider presenting some results about this.

Author Response

Thanks a lot for your kind suggestion!

The response can be found in the attached file.

Round 2

Reviewer 1 Report

The authors responded to all my comments.

The title does not reflect the subject covered in the article. Please do not change the title proposed in the first version.

Reviewer 2 Report

Thank you for revision!